# Evaluation of the Toxic Activity of the Graphene Oxide in the Ex Vivo Model of Human PBMC Infection with *Mycobacterium tuberculosis*

**DOI:** 10.3390/microorganisms11030554

**Published:** 2023-02-22

**Authors:** Alessandro Salustri, Flavio De Maio, Valentina Palmieri, Giulia Santarelli, Ivana Palucci, Delia Mercedes Bianco, Federica Marchionni, Silvia Bellesi, Gabriele Ciasca, Giordano Perini, Maurizio Sanguinetti, Michela Sali, Massimiliano Papi, Marco De Spirito, Giovanni Delogu

**Affiliations:** 1Dipartimento di Scienze Biotecnologiche di Base, Cliniche Intensivologiche e Perioperatorie—Sezione di Microbiologia, Università Cattolica del Sacro Cuore, 00168 Rome, Italy; 2Dipartimento di Scienze di Laboratorio e Infettivologiche, Fondazione Policlinico Universitario “A. Gemelli”, IRCCS, 00168 Rome, Italy; 3Istituto dei Sistemi Complessi, CNR, 00168 Rome, Italy; 4Fondazione Policlinico Universitario “A. Gemelli”, IRCSS, 00168 Rome, Italy; 5Dipartimento di Diagnostica per Immagini, Radioterapia Oncologica ed Ematologia, Fondazione Policlinico Universitario “A. Gemelli”, IRCCS, 00168 Rome, Italy; 6Dipartimento di Neuroscienze, Università Cattolica del Sacro Cuore, 00168 Rome, Italy; 7Mater Olbia Hospital, 07026 Olbia, Italy

**Keywords:** graphene oxide, mycobacterium tuberculosis infection, PBMCs

## Abstract

Graphene Oxide has been proposed as a potential adjuvant to develop improved anti-TB treatment, thanks to its activity in entrapping mycobacteria in the extracellular compartment limiting their entry in macrophages. Indeed, when administered together with linezolid, Graphene Oxide significantly enhanced bacterial killing due to the increased production of Reactive Oxygen Species. In this work, we evaluated Graphene Oxide toxicity and its anti-mycobacterial activity on human peripheral blood mononuclear cells. Our data show that Graphene Oxide, different to what is observed in macrophages, does not support the clearance of *Mycobacterium tuberculosis* in human immune primary cells, probably due to the toxic effects of the nano-material on monocytes and CD4+ lymphocytes, which we measured by cytometry. These findings highlight the need to test GO and other carbon-based nanomaterials in relevant in vitro models to assess the cytotoxic activity while measuring antimicrobial potential.

## 1. Introduction

Tuberculosis (TB), a disease caused by *Mycobacterium tuberculosis* (Mtb), is one of the main causes of death from a single infectious agent [1]. Tuberculosis is often curable with a standardized therapeutic regimen. The latest WHO guidelines strongly recommend a 6-month regimen of isoniazid (H), rifampicin (R), ethambutol (E) and pyrazinamide (Z) for drug-susceptible TB (both pulmonary and extra-pulmonary): all four drugs for the first 2 months, followed by H and R for the remaining 4 months [2]. Tuberculosis disease caused by multi-drug-resistant (MDR-TB) and extensively drug-resistant (XDR-TB) Mtb strains requires longer and more complex regimens with lower therapeutic success rates [3,4]. Current efforts to improve anti-TB regimens mainly aim at enhancing activity against MDR-TB and XDR-TB and new and out-of-the-box approaches are needed to address these challenges [5].

Recently, Carbon-based Nanomaterials (CNM), such as fullerenes, nanotubes, diamonds, graphite, graphene and its conjugate Graphene Oxide (GO), have shown a broad direct antibacterial effect in in vitro assays [6,7,8,9,10], including activity against Mtb and other nontuberculous mycobacteria [11,12,13,14]. Surface-charged fullerenes can inhibit the growth of *M. avium* and Mtb [11], GO in a reduced state exerts antibacterial effect against Mtb [13] and GO-Ethambutol particles inhibited *M. smegmatis* growth in axenic liquid culture [14,15]. It shall be noted that in many of these experiments functionalized forms rather than a “pure form” of GO were used, with the functional groups providing peculiar physical and chemical features [16]. Interestingly, the investigators focused their studies mainly at evaluating the activity of these CNM on bacteria, paying less attention to the effect of these molecules on eukaryotic cells.

In our previous works, we investigated the GO effect on mycobacteria, including Mtb, both in axenic culture and in a macrophage infection model. Graphene Oxide did not show a direct bactericidal activity, but it was able to entrap mycobacteria in a net interfering with the normal infection of macrophages [17]. Interestingly, the co-administration of GO with the second-line drug linezolid resulted in a synergistic anti-Mtb effect, also due to an increased production of reactive oxygen species (ROS) [18]. However, further studies are needed to assess the therapeutic potential of the proposed combined approach. To this end, we designed an experimental setup based on the infection of Peripheral Blood Mononuclear Cells (hPBMCs) with Mtb, which offers a reliable model to measure the activity against Mtb while assessing any toxicity on the multiple types of circulating cells that contribute to contain the infection.

## 2. Materials and Methods

### 2.1. Graphene Oxide Characterization

Graphene Oxide (GO) water dispersion at a concentration of 4 mg/mL was purchased from Graphenea (San Sebastián, Spain). For GO characterization, 100 μL of sample (10 µg/mL) was deposited on sterile mica slides and air-dried overnight for atomic force microscopy imaging (AFM) with a NanoWizard II (JPK Instruments AG, Berlin, Germany) [15]. The images were acquired using silicon cantilevers with high aspect-ratio conical silicon tips (CSC36 Mikro-Masch, Tallinn, Estonia) and characterized by an end radius of about 10 nm, a half conical angle of 20° and a spring constant of 0.6 N/m. Small scan areas (6 × 6 μm) were imaged. Appendix A depicts a representative image of GO sheets from the diluted water dispersion. Samples had a hydrodynamic radius ranging from 500 to 900 nm and an average height of 1 nm, indicating single-layered GO sheets in solution, in accordance with our previous findings. Full characterization of the GO is reported elsewhere [17,18].

### 2.2. Bacterial Manipulation

Each experiment was performed using the Mtb reference strain (Mtb H37Rv). Bacteria were grown in 7H9 broth medium (Difco) enriched with 10% albumin dextrose catalase (ADC) (Sigma-Aldrich) and 0.05% Tween 80 (Sigma-Aldrich, Schnelldorf, Germany) without antibiotics, at 37 °C and 110 rpm agitation. When bacterial culture reached an OD_600_ ~ 0.6, it was supplemented with 20% of sterile pure glycerol (Carlo Erba Reagents, Cornaredo, Italy) and aliquots were stored at −80 °C [19]. All experiments that involved Mtb manipulation were performed in a Biosafety level 3 laboratory (BSL3) in the Institute of Microbiology of IRCCS—Fondazione Policlinico Gemelli.

### 2.3. hPBMC Isolation

Human Peripheral Blood Mononuclear Cells (hPBMCs) were isolated from buffy coats of healthy donors [20]. Blood was diluted with sterile Phosphate Buffered Saline (PBS) (Euroclone, Pero, Italy) and gently poured into a tube containing Ficoll human Lympholyte^®^ (CEDARLANE, Burlington, Ontario, Canada), and finally centrifuged at 1500 RPM for 30 min at room temperature (23 °C) with no brake, as per manufacturer’s instructions. Lymphocytes and monocytes were collected and washed with PBS. Finally, cells were re-suspended in Roswell Park Memorial Institute (RPMI) 1640 culture medium (Euroclone) enriched with 10% foetal bovine serum (Corning), 1% Glutamine (Euroclone), 1% Pyruvate (Euroclone) and plated in 48-wells plate (NEST) at a final concentration of 1.2 × 10^6^ cells/mL [21].

### 2.4. hPBMC In Vitro Infection Model

A Multiplicity of Infection (MOI) of 1 with respect to monocytes (corresponding to ~5% of the hPBMCs, ~6 × 10^4^ cells) was used to infect hPBMCs. To verify the synergistic effect of GO and linezolid (LZD), infected hPBMCs were immediately treated with different LZD concentrations (0.25, 0.5 and 1 µg/m) and with 250 µg/mL GO + LZD at the same concentrations [18] (Figure 1A).

To further measure the ability of GO against Mtb ongoing infection treatment, infected hPBMCs were infected and Mtb let multiply in absence of any treatment for four days; then, treatment with LZD alone (0.25, 0.5 and 1 µg/m) or in combination with 250 µg/mL GO was started (Figure 1C). Bacterial survival was evaluated at 7 days post-infection by assessing CFU counting [21]. Briefly, cell culture medium was discharged, and cell layers were lysed with a 0.01% Triton X-100. The obtained suspensions were diluted in phosphate buffer saline (PBS) containing 0.05% Tween 80 (Sigma-Aldrich, Germany) and plated on mycobacterial Middlebrock 7H11 solid medium (BD Difco^TM^) enriched with 10% Oleic-Acid, Albumin Dextrose Catalase (OADC). Peripheral Blood Mononuclear Cells were also treated immediately or four days post-Mtb infection with GO alone at 1, 10 or 100 µg/mL and bacterial survival evaluated as previously described.

Moreover, hPBMCs pre-stimulated either with Mtb H37Rv (MOI 1:10.000), 1 µg/mL of tuberculous Purified Protein Derivative (PPD), or Phorbol Myristate acetate (PMA) at final concentration of 50 ng/mL, were infected simultaneously with Hygromycin B-resistant Mtb (Mtb^Hyg^) and simultaneously treated with GO at final concentrations of 10 and 100 µg/mL. Bacterial survival was evaluated 7 days later, assessing CFUs by plating on 7H11 solid medium (BD Difco^TM^) enriched with 50 µg/mL Hygromycin B.

### 2.5. GO Toxicity Evaluation on hPBMCs

Peripheral Blood Mononuclear Cells were treated with GO (10 and 100 µg/mL) for 1 h to evaluate cytotoxicity and untreated hPBMCs were included as control. Cell populations were identified using a combination of 8 fluorochrome-labelled monoclonal antibodies: HLA-DR V450 (clone L243), CD45 BV500 (clone HI30), CD64 FITC (clone 10.1), CD4 PE (clone RPA-T4), 7-AAD PerCP-Cy5.5 (viability marker), CD3 APC-H7 (clone SK7) (BD Biosciences), CD8 PC7 (clone SFCI21Thy2D3) and CD25 APC (clone B1.49.9) (Beckman Coulter). An aliquot from each suspension was incubated at room temperature with the above-mentioned antibodies for 20 min. Cells were washed with PBS containing 1% bovine serum albumin (BSA) by centrifuging at 1500 RPM for 5 min and finally resuspended in PBS. Data were acquired using the Cytoflex cytofluorimeter and analysed using Kaluza software (Beckman Coulter). Data analysis was carried out following a gating strategy composed of several and serial steps on at least 20,000 events.

7AAD PC5.5-A versus Side Scatter (SSC-A), Forward Scatter Area (FSC-A) versus SSC-A and Forward scatter Height (FSC-H) versus FSC-A gatings were achieved to distinguish live cells from cell debris, artifacts and GO aggregates. Following gatings were performed to distinguish different cell populations: (a) CD3 APC-A750-A versus SSC-A was used to select lymphocyte population; (b) CD8 PC7-A versus CD4 PE-A was used to distinguish CD8 from CD4 lymphocyte populations; (c) a CD64 FITC-A versus SSC-A was used to evidence monocytes. Cell activation and maturation rates were analysed using DR PB450-A versus CD25 APC-A and CD64 FITC-A versus DR PB450-A was used to evaluate lymphocyte and monocyte activation, respectively.

Data acquired by suspensions of GO at 10 and 100 µg/mL in RPMI were acquired as auto-fluorescence control. An aliquot of each GO suspension was labelled with 7-AAD PerCP-Cy5.5 monoclonal antibody and analysed as previously described.

### 2.6. Statistical Analysis

*Microsoft Excel* (2016) and *GraphPad Prism* software version 9.0.0 (GraphPad software) were used for data collection and analysis. All data were expressed as mean plus SD and analysed by one-way or two-way ANOVA followed by appropriate corrections.

## 3. Results

### 3.1. Combined Administration of GO with LZD Does Not Enhance the Anti-Mtb Activity of hPBMC

Our previous work demonstrated that the co-administration of GO and LZD enhanced the activity of LZD against Mtb cultured in axenic culture and in a murine macrophages infection model [18]. To further investigate the potential of GO as an adjunctive therapy, we tested the activity of GO and of the GO/LZD combination in Mtb-infected human hPBMCs. Peripheral Blood Mononuclear Cells obtained from healthy donors were infected with Mtb H37Rv and then treated with GO (250 µg/mL) and with LZD (0.25, 0.5 and 1 µg/mL) alone or in combination with GO, as shown in Figure 1. These formulations were administered simultaneously with Mtb infection (co-treatment, Figure 1A) or 4 days post-infection (post-treatment, Figure 1C) and bacterial survival was assessed by CFUs counting at day 7 post-infection.

When LZD was administered at MIC concentration (1 µg/mL), a significant reduction in Mtb survival was observed in both co-treatment and post-treatment. Conversely, treatment with GO or GO with LZD at different concentrations (0.25, 0.5 and 1 µg/mL) did not impair Mtb replication but rather resulted in CFUs comparable to control (post-treatment) or higher (co-treatment), suggesting that treatment with GO negatively affects the ability of hPBMCs to contain Mtb replication (Figure 1B,D). Lower GO concentrations (1, 10 and 100 µg/mL) (Figure 2) did not result in any reduction in Mtb CFUs regardless of the time of treatment, during (Figure 2A) or after (Figure 2C) Mtb infection.

Taken together, these findings suggest that, differently to what was previously observed in murine macrophages [17,18], either GO exerts a toxic effect on hPBMCs or GO loses its activity when administered at low concentrations, resulting in the failure to contain Mtb infection in this experimental model.

### 3.2. hPBMCs Incubated with GO Show a Reduction of Not Differentiated Monocytes

There are conflicting results on the toxicity of GO on hPBMCs, with some studies indicating a clear toxic effect and others showing no impact on hPBMC viability and proliferation potential [12,22,23]. To investigate the effect of GO on hPBMCs, we used flow cytometry analysis to pinpoint the viability of the different cellular subsets. Peripheral Blood Mononuclear Cells were incubated with GO (10 and 100 µg/mL) and one hour later cell viability was assessed measuring the 7AAD PC5.5-A signal (Figure 3A–C). Positive 7AAD PC5.5-A cells indicating cell death and viability were reported as percentage of total events (cells). Debris and doublets were removed from the 7AAD+ cell population by using Forward scatter Height (FSC-H) versus Forward Scatter Area (FSC-A) gating. As shown in Figure 3D–H, the treatment of hPBMCs with GO significantly reduced cell viability in a dose-dependent manner.

Lymphocyte (CD3 positive cells distinguished among CD4 and CD8 positive cells) and monocyte (CD64 positive population) viability was further evaluated separately (Figure 4A–I). Whereas treatments with GO at 100 µg/mL slightly impaired CD4+ lymphocytes, no significant effect was observed on the CD8+ component. Conversely, both concentrations of GO significantly affected monocyte viability (Figure 4J).

Finally, we assessed cell activation through GO stimulation by measuring the markers CD25 (early activation) and DR (late activation) [24,25], as reported in Figure 5A–C,E–G, representing CD4+ and CD8* lymphocytes, respectively. Graphene Oxide exposure did not prompt the massive activation of CD4+ (Figure 5D) and CD8+ lymphocytes (Figure 5H). On the other side, monocyte activation was evaluated via DR exposure measurement (Figure 5I–K), which is down-regulated when monocytes maturate towards a differentiated phenotype. Despite the significant loss of viable monocytes after GO treatment, monocytes showed no sign of activation (Figure 5L).

Taken together, our results showed that GO affects the viability of human immune cells, mainly monocytes and, to a lesser extent, CD4 lymphocytes. Moreover, it prompts an early activation phenotype in CD4 lymphocytes.

### 3.3. GO Exerts a Generalized Toxic Effect on Human Immune Cells

To investigate whether GO affects exclusively the viability of undifferentiated cells, or also mature macrophages, we pre-treated hPBMCs to enhance monocyte differentiation before GO treatment. Peripheral Blood Mononuclear Cells were stimulated either with Mtb H37Rv (MOI of 1:10,000), tuberculous Purified Protein Derivative (PPD) or Phorbol 12-Myristate 13-Acetate (PMA) and three-day post stimulation hPBMCs were infected with a Hygromycin-resistant Mtb recombinant strain (Mtb^Hyg^). Bacterial survival was evaluated 7 days post-infection. As shown in Figure 6b, bacterial growth was higher in GO-treated samples. Furthermore, Mtb H37Rv-, PPD- or PMA-stimulated cells before GO treatment did not show an improvement of antimycobacterial activity. This result confirms that GO, at least in this formulation, also exerts a generalized toxic effect on previously triggered immune cells.

## 4. Discussion

The medical application of CNM, and particularly of GO, has been a hot topic in the last ten years, mostly due to its biocompatibility and antimicrobial properties [26]. When administered in vivo, GO preferentially accumulates in the lungs, prompting many to investigate its potential use as adjunctive therapy in respiratory infections such as TB [6,27,28]. However, the use of GO remains controversial mainly due to the variable cytotoxicity observed that can vary depending on the GO physico-chemical features (lateral size, impurities due to manufacturing methods, concentration and functionalization) and the experimental model used [22,29,30].

We have demonstrated that GO can entrap extracellular mycobacteria during the infection of murine macrophages [17] and, thanks to surface-exposed functional groups, enhances the anti-Mtb activity of linezolid in both axenic culture and in infected macrophages [18]. Hence, we aimed to investigate GO activity in a relevant ex vivo model as hPBMC infection with Mtb. Given the presence of multiple cell types, this model offers a good representation of the Mtb–host interaction [20,31]. Different to what we previously observed in Mtb-infected macrophages, GO administration alone or in combination with linezolid exerted a detrimental effect on the ability of hPBMCs to contain Mtb infection, with an unexpected increase in the overall microbial burden assessed by CFU counting, when compared to untreated cells. Flow cytometry analysis on hPBMCs indicates that GO has a cytotoxic effect on immune cells, with a pronounced toxicity on monocytes and partially on CD4+ T lymphocytes, suggesting that in this experimental model GO treatment fails to contain Mtb due to the damage exerted on key cell types involved in the control of Mtb [32,33].

These results support previous findings regarding GO-COOH toxicity on CD4 lymphocytes, a compound similar to the GO form used in our experiments but with a hydrodynamic radius of ~180 nm [34]. It must be noted that although nanomaterials with a lateral size shorter than 1 μm are defined as moderately toxic, the maximum peak of “toxic GO” showed an average length of 100–200 nm [35], a smaller size compared to the GO form used in this work. However, GO heterogeneity in terms of lateral size could be relevant to identify the nontoxic concentrations.

It is also important to highlight that in our experimental settings hPBMCs were incubated with GO for 4 to 7 days, while in our previous study the incubation time was much shorter (only 4–24 h) [17,18]. In fact, GO-induced ROS production is dependent on exposure time [36], suggesting that longer exposition time to this form of GO may affect hPBMC viability and function, thus impairing the overall anti-mycobacterial response.

Intriguingly, exposure of LPS-activated primary Human Monocytes Derived Macrophages to GO triggers a robust secretion of pro-inflammatory cytokines, while this effect is much less pronounced in non-LPS-activated macrophages [37]. The GO-induced secretion of cytokines in LPS-activated macrophages is mediated by a massive production of ROS that in turn activates the inflammasome pathway, driving cells to apoptosis [37,38]. It has also been shown that exposure to GO prompts the early stimulation of lymphocyte proliferation that then triggers apoptosis due to oxidative stress [35,39,40,41]. Hence, exposure to GO of immune cells in an activated state, as can be the case in hPBMCs, may affect their viability and functional status. Given that GO has such a deleterious effect on monocytes and lymphocytes and considering the fundamental role of these cells in Mtb infection, the toxic effect observed on these immune cells may explain the failure in controlling Mtb infection in our experiments. However, the molecular and immunological mechanism responsible for the deleterious effects on these cells remains to be determined. While the use of a reduced and PEGylated form of GO in glioblastoma cells did not trigger the STING pathway [42], more studies are needed to elucidate the impact of GO on key intracellular pathways.

Mutations and chromosomal abnormalities, resulting from multiple replications in immortalized cells, could determine resistance to environmental stresses, including GO [43,44,45]. The observation of a greater and diverse susceptibility of primary cells, especially monocytes, to GO, compared to immortalized cell lines, suggests that the results obtained with cell lines are not necessarily always representative of what happens in ex vivo and in vivo systems, suggesting the need of common standardized models.

## 5. Conclusions

The data shown in the present study prompt some concern for the use of GO in vivo given the toxicity displayed toward primary human immune cells. These findings are somewhat disappointing given the potential of GO-based therapy against mycobacterial and other bacterial infections observed in previous studies [6,8,46]. However, the functionalization of GO with molecules and chemical species does impact on the biological properties and may eventually reduce toxicity. The characterization of these functionalized GO shall include the proper evaluation of toxicity in relevant ex vivo models, so as to identify the combinations showing adequate biocompatibility while maintaining anti-microbial properties. Moreover, other GO-derived carbon-based nanomaterials such as for instance Quantum dots [6,47,48,49] may provide more effective and less toxic scaffolds and overcome the observed limitations of the tested GO.

## Figures and Tables

**Figure 1 microorganisms-11-00554-f001:**
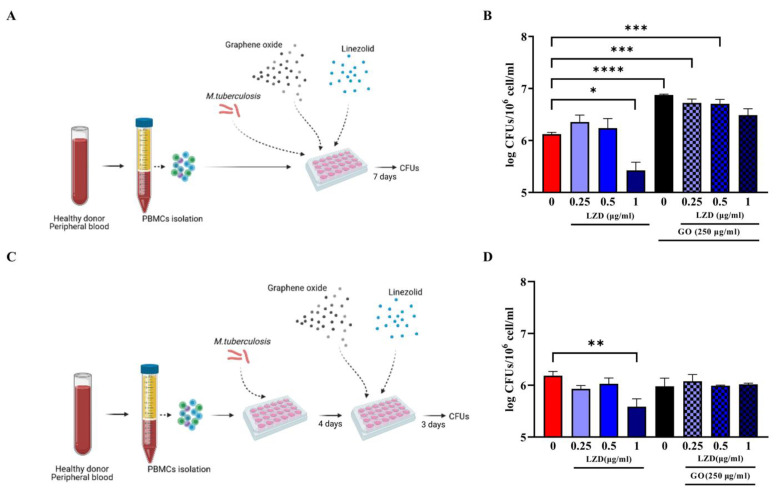
**Effect of GO-LZD combination during Mtb in ex vivo hPBMC infection model.** The GO-LZD effect on *Mycobacterium tuberculosis* survival was evaluated in ex vivo human PBMC infection model. Graphene Oxide at a final concentration of 250 µg/mL was administered alone or in combination with LZD at a final concentration of 0.25, 0.5 and 1 µg/mL. Peripheral Blood Mononuclear Cells were infected with 6 × 10^4^ CFUs of Mtb H37Rv strain (Multiplicity Of Infection, MOI 1:1 based on monocyte percentages) and GO or GO+LZD solutions were administered as co-infection treatment or as 4 days post-infection treatment. Schematic representations of experimental setting (**A**,**C**); Bacterial survival was evaluated 7 days post-infection through CFU count and are represented in logarithmic scale (**B**,**D**). Each experiment was repeated three times. When GO was added as co-infection treatment, there was a decrease in CFU count at 7 days post-infection only with LZD 1 µg/mL (MIC concentration) with respect to untreated cells (not significant), while when GO alone was added, or in combination with LZD 0.25, 0.5 and 1 µg/mL there was a higher bacterial replication (**B**). In post-infection treatment, only when LZD at MIC point was added there was a decrease in CFU count. With GO alone or in combination with LZD, bacterial load was comparable with control (**D**). (* *p* value < 0.05; ** *p* value < 0.01; *** *p* value < 0.001, **** *p* value < 0.0001).

**Figure 2 microorganisms-11-00554-f002:**
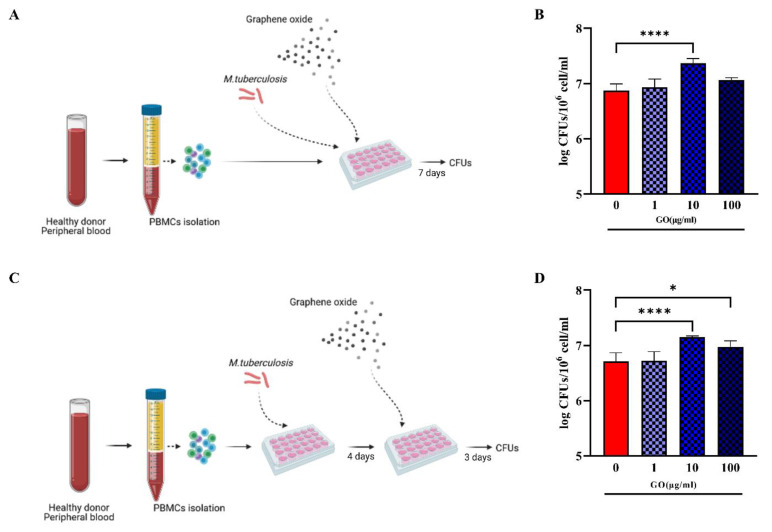
**Effect of GO during Mtb ex vivo hPBMCs infection model.** To confirm GO impairment of infection control, hPBMC infection models in Figure 1 were repeated with GO alone at 1, 10 and 100 µg/mL, without LZD (**A**,**C**). Bacterial survival was evaluated through CFU count and represented in logarithmic scale (**B**,**D**). When GO was added as co-infection treatment, there was an increase in CFU count (not significant for 1 and 100 µg/mL and *p* value < 0.0001 for GO 10 µg/mL). When it was added as post-treatment, there was still an uncontrolled bacterial growth (*p* values, respectively, =0.0270 for GO 10 µg/mL and <0.0001 for GO 100 µg/mL). (* *p* value < 0.05; **** *p* value < 0.0001).

**Figure 3 microorganisms-11-00554-f003:**
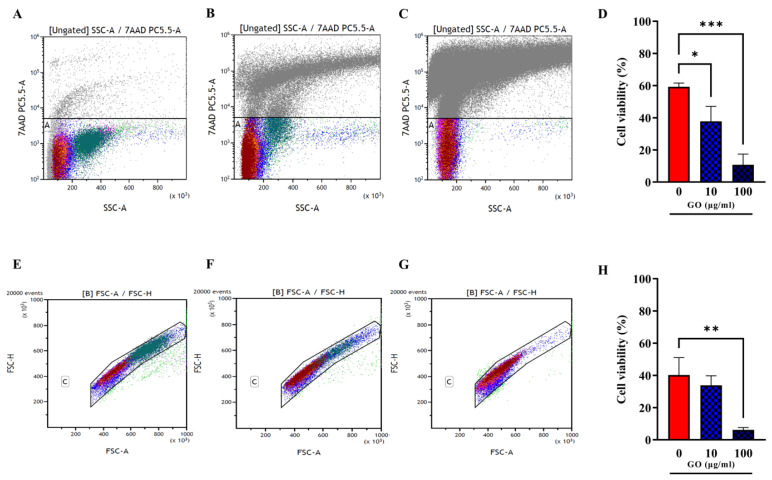
**Viability assessment of hPBMCs after GO treatment.** Total cell viability evaluated through 7AAD PC5.5-A versus Side Scatter (SSC-A) gating was evaluated among not treated hPBMCs (**A**) and hPBMCs following incubation for 1 h, with GO at the final concentrations of 10 µg/mL (**B**) and 100 µg/mL (**C**). (**A**–**C**) Charts were quantitatively analysed and represented as histograms showing average ± standard deviation. (**D**) Cells treated with GO 100 µg/mL show a strong reduction in viability compared to untreated (*p* value = 0.0002), while cells treated with GO 10 µg/mL undergo a slighter reduction in viability. Cell viability was analysed purging samples from doublets and debris ((**E**); not treated hPBMCs; (**F**,**G**) cells incubated with GO at 10 µg/mL and 100 µg/mL) by using Forward scatter Height (FSC-H) versus Forward Scatter Area (FSC-A) gating. (**H**) Quantitative representation of (**E**–**G**) panels. Cell viability decrease is confirmed for GO 100 µg/mL (*p* value = 0.0089). (* *p* value < 0.05; ** *p* value < 0.01; *** *p* value < 0.001).

**Figure 4 microorganisms-11-00554-f004:**
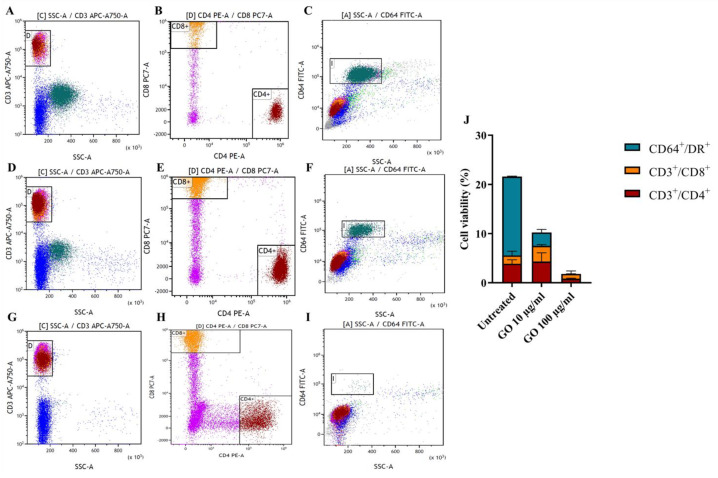
**Viability assessment of hPBMC subpopulations after GO treatment.** Single lymphocyte populations (CD3+/CD8+ and CD3/CD4+) and monocyte populations (CD64+) were evaluated among untreated cells (**A**–**C**) and following incubation for 1 h, with GO at the final concentrations of 10 µg/mL (**D**–**F**) and 100 µg/mL (**G**–**I**). Lymphocyte viability was evaluated through CD3 APC-A750-A versus Side scatter (SSC-A) gating (**A**,**D**,**G**) and CD4+ and CD8+ populations were distinguished from each other through CD8 PC7-A versus CD4 PE-A gating (**B**,**E**,**H**). Monocyte population was emphasized through CD64 FITC-A versus Side scatter SSC-A gating monocyte population (**C**,**F**,**I**). Effects of GO on different populations were quantitatively represented as histograms showing average ± standard deviation and based on total percentages (**J**). Graphene Oxide 100 µg/mL exerts a high toxic effect on monocytes (*p* value < 0.0001) and CD4^+^ lymphocytes (*p* value = 0.0077) (**G**,**H**,**I**) compared to untreated cells (**A**–**C**), rather than on CD8^+^ cells. Graphene Oxide 10 µg/mL (**D**–**F**) seems to exclusively affect monocytes (*p* value < 0.0001).

**Figure 5 microorganisms-11-00554-f005:**
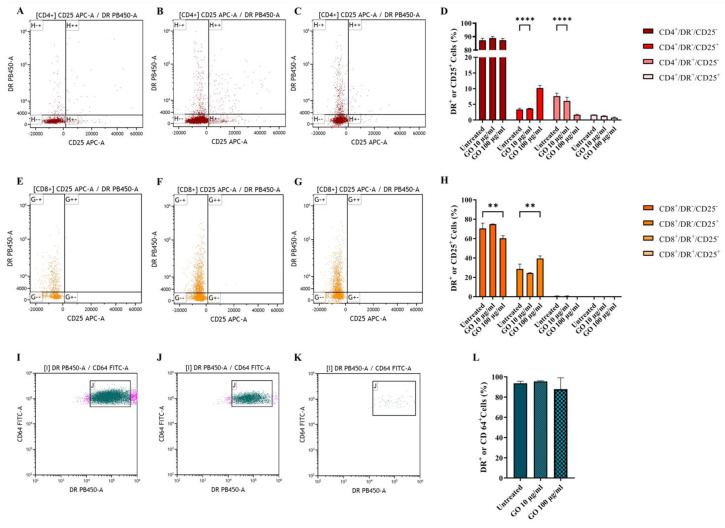
**Graphene Oxide effect on cell activation.** Activation of CD4+ (**A**–**C**), CD8+ (**E**–**G**) and CD64+ (**I**–**K**) populations was evaluated after 1 h incubation. Untreated cells (**A**,**E**,**I**), GO 10 µg/mL treated cells (**B**,**F**,**I**) and GO 100 µg/mL (**C**,**G**,**K**) after 1 h incubation. CD4+ activation was evaluated through DR PB450-A versus CD25 APC-A gating on CD4+ cells (**A**–**C**). CD8+ activation was evaluated through DR PB450-A versus CD25 APC-A gating on CD8+ cells (**E**–**G**). Monocyte activation was evaluated through CD64 FITC-A versus Side scatter SSC-A gating on CD64+ cells (**I**–**K**). Activation of cells based on parental percentages is quantitatively represented as histograms showing average ± standard deviation (**D**,**H**,**L**). The presence of nano-material does not affect cellular activation since, when exposed to GO, CD4+ and CD8+, cells show only an early activation phenotype, comparable with untreated cells (*p* value, respectively, <0.0001 and =0.0014) while CD64+ alive cells show no sign of maturation. (** *p* value < 0.01; **** *p* value < 0.0001).

**Figure 6 microorganisms-11-00554-f006:**
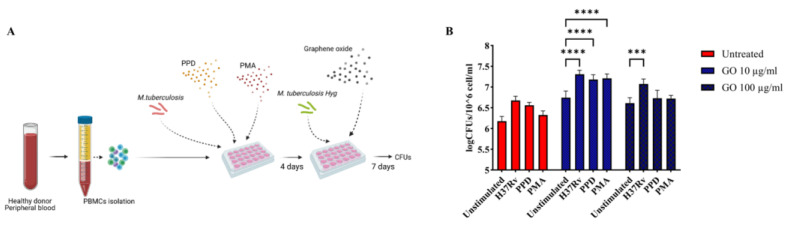
**Graphene Oxide toxicity affects not only undifferentiated cells, but also mature macrophages.** Graphene Oxide effect on *Mycobacterium tuberculosis* survival was evaluated in ex vivo pre-stimulated human PBMC infection model to determine if GO toxicity is restricted to undifferentiated monocytes. Peripheral Blood Mononuclear Cells were pre-stimulated with low doses of H37Rv, PPD and PMA as described in Materials and Methods section to trigger monocyte maturation. Four- day post-stimuli cells were infected with H37Rv resistant to Hygromycin B at the same MOI of previous experiments and co-treated with GO 10 and 100 µg/mL as schematically represented in (**A**). Bacterial survival was evaluated 7 days post-infection through CFUs (logarithmic scale) (**B**). The experiment was repeated three times. When hPBMCs were stimulated with Mtb H37Rv and treated with GO 100 µg/mL, there was an increment in bacterial survival (*p* value = 0.0003). When hPBMCs were treated with GO 10 µg/mL, there was an increment of CFUs independently of stimuli (*p* values < 0.0001). The GO exerts its toxic effect also on mature cells. (*** *p* value < 0.001, **** *p* value < 0.0001).

## Data Availability

Data are available upon request to the corresponding author.

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
