# Peer review of "Evaluation of the Toxic Activity of the Graphene Oxide in the Ex Vivo Model of Human PBMC Infection with Mycobacterium tuberculosis"

_microorganisms, 2023, doi:10.3390/microorganisms11030554_

Round 1
Reviewer 1 Report
Dear Authors,
Please find attached the comments.
Best wishes

Author Response
Reviewers' comments:
Reviewer 1
Tuberculosis (TB) is one of the top 10 causes of death worldwide. This scenario is further complicated by the insurgence of multidrug-resistant and extensively drug-resistant TB. The identification of appropriate drugs with multi-target affinity profiles is considered to be a widely accepted strategy to overcome the rapid development of resistance.
In this paper, Salustri A et al., performed experiments to evaluate the potential of toxic activity of the Graphene Oxide (GO) in the ex vivo model of human peripheral blood mononuclear cells (hPBMCs) infection with Mycobacterium tuberculosis (Mtb).
Authors written an interesting scientific article, the experimental design, data analysis, and conclusions are appropriate for the study for the most part.
Their previous study showed that GO did not show a direct bactericidal activity, but it was able to entrap mycobacteria in a net interfering with the normal infection of macrophages. However, interestingly, co- administration of GO with the second-line drug linezolid resulted in a synergistic anti-Mtb effect, also due to an increased production of reactive oxygen species (ROS). Following this research line, in this study, the researchers, designed an experimental setup based on the infection of hPBMCs with Mtb, which surely offers a reliable model to measure the activity against Mtb while assessing any toxicity on the multiple types of circulating cells that contribute to contain the infection.
Authors’ results showed that differently to what had previously observed in murine macrophages, either GO exerts a toxic effect on hPBMCs or that GO loses its activity when administered at low concentrations, resulting in the failure to contain Mtb infection in this experimental model. Flow cytometry analysis on hPBMCs reported that GO has a cytotoxic effect on immune cells, with a pronounced toxicity on monocytes and partially on CD4+ T lymphocytes, suggesting that in this experimental model GO treatment fails to contain Mtb due to the damage exerted on key cell types involved in the control of Mtb.
Considered that there are conflicting results on the toxicity of GO on hPBMCs will be necessary perform other further studies in order to have more data for a correct statistical analysis. Other than that, it is necessary the functionalization of GO with molecules and chemical species in order to reduce toxicity.
Considering as above reported, I have just a curiosity:
Are there studies on the biological effects of GO (or other medical application of Carbon-Based Nanomaterials) in human immune cells co-infected with Mtb and HIV? In the discussion's chapter, authors are invited to explain the biological effects of GO arguing with more details the molecular mechanisms related to the immune response signalling pathways with particular attention on the STING signaling pathway, Toll-like receptor (TLR) signalling and Inflammasome.
Regards
We thank the reviewer for the overall positive comments on the whole manuscript.
Unfortunately, data and on the interaction of GO with human cells infected with HIV, or previously “stimulated with Mtb”, or with another relevant infectious agent, are limited and not sufficient to provide a satisfactory immunological and molecular framework for a better understanding of the mechanisms involved.
Regarding the biological effects and molecular mechanisms related to the GO-dependent toxic activity on specific immune cells, we revised extensively the discussion section and introduced a new paragraph, with a link to the STING pathway as suggested by the reviewers. We are aware of the complexity and for this reason we used hPBMC obtained from healthy volunteers to minimize variability due to previous infections.

Reviewer 2 Report
The presented paper for review is very interesting, addressing important research problems. Nevertheless, in addition to a few editorial errors, I have a few comments.
Editorial:
1. in the text Authors have different font of writing
2. it could be space in this: 6x6 (6 x 6) and in some places is to much space
3. in my opinion figures could be in the “Results” not on the end of the paper.
4. References! in the text we name of the authors, date etc, but in the part “References” they have as numbers… it is not ok. It could be the same, so it must be done different: (1) one posibilities is as numerous in the text and compatible numerous in the References or (2) as names in the text and in the REFERENCES in alphabetical order. It depend on the journal.
Other comments:
5. The first paragraph of the “Results” in my opinion it is material and methods.
6. in this part “Lymphocytes (CD3 positive cells distinguished among CD4 and CD8 positive cells)
and monocytes (CD64 positive population) viability was further evaluated separately
(Figure 4A-I). Whereas treatments with GO at 100 μg/mL slightly impaired CD4+
lymphocytes, no significant effect was observed on the CD8+ component. Conversely,
both concentrations of GO dramatically affected monocytes viability (Figure 4J).” I don’t know what’s mean dramatically?
7. “Discusion” in my opinion this discussion is very general and not really specific to the results of the study. The conclusions are very general and take into account what was used in the experiments, i.e. different concentrations of GO, different exposure times and others.
8. “The data shown in the present study prompts some concern on the use of GO in vivo
given the toxicity displayed toward primary human immune cells. These findings are
somehow disappointing given the potential of GO-based therapy against mycobacterial
and other bacterial infections observed in previous studies (Bugli et al., 2018; Tudose et
al., 2019). However, functionalization of GO with molecules and chemical species do
impact on the biological properties and may eventually reduce toxicity. Characterization
of these functionalized GO shall include proper evaluation of the toxicity in relevant ex
vivo model, so to identify the combinations showing adequate biocompatibility while
maintaining the anti-microbial properties. Moreover, other GO-derived carbon-based
nanomaterials such as for instance Quantum dots (De Maio, Palmieri, De Spirito, et al.,
2019; Deng et al., 2022; Raja et al., 2022; Vasyukova et al., 2022) may provide more effective
and less toxic scaffolds and overcome the observed limitations of the tested GO.” It could be as Conclusion.
9. Figure 5. I don't see different between A ,B C picture, because description is the same , in text (results) we don't have any information it could be maturation in time or in what ? I don't understand part of this figure. The same situation is between E-G and I-K. May be it is my little experience in the similar experiments. And id the description of this Figure: (D, H, I) I is a mistake should be L.
Author Response
Reviewer 2
The presented paper for review is very interesting, addressing important research problems. Nevertheless, in addition to a few editorial errors, I have a few comments.
We thank the reviewer for the overall positive comments on the whole manuscript. In the point-by-point rebuttal, we will address the specific points raised by the reviewer.
Editorial:
- In the text Authors have different font of writing
We thank the reviewer for the observation. We revised the paper to correct the font.
- It could be space in this: 6x6 (6 x 6) and in some places is too much space
We thank the reviewer for this observation. We revised the paper to correct the space issues.
- In my opinion figures could be in the “Results” not on the end of the paper.
We thank the reviewer for the observation. We are submitting a revised version of the manuscript in the correct template.
- References! In the text we name of the authors, date etc, but in the part “References” they have as numbers… it is not ok. It could be the same, so it must be done different: (1) one possibilities is as numerous in the text and compatible numerous in the References or (2) as names in the text and in the REFERENCES in alphabetical order. It depend on the journal.
We apologize for the mistakes, and we appreciate reviewer’s suggestions. In the revised manuscript, we have corrected the style for the references.
Other comments:
- The first paragraph of the “Results” in my opinion it is material and methods.
We thank the reviewer for the comment. While we understand that the first paragraph of the result section is certainly introductory, we think that it would help the reader to have a summary on the rationale and the experimental settings used.
- In this part “Lymphocytes (CD3 positive cells distinguished among CD4 and CD8 positive cells) and monocytes (CD64 positive population) viability was further evaluated separately (Figure 4A-I). Whereas treatments with GO at 100 μg/mL slightly impaired CD4+ lymphocytes, no significant effect was observed on the CD8+ component. Conversely, both concentrations of GO dramatically affected monocytes viability (Figure 4J).” I don’t know what’s mean dramatically?
We thank the reviewer for the suggestion, and we have changed the term in “significantly”.
- “Discussion” in my opinion this discussion is very general and not really specific to the results of the study. The conclusions are very general and take into account what was used in the experiments, i.e. different concentrations of GO, different exposure times and others.
We thank the reviewer for the observation. In the revised manuscript, we have improved the discussion regarding the effects due to GO treatment and Mtb replication failure. Furthermore, a conclusion section has been included.
- “The data shown in the present study prompts some concern on the use of GO in vivo given the toxicity displayed toward primary human immune cells. These findings are somehow disappointing given the potential of GO-based therapy against mycobacterial and other bacterial infections observed in previous studies (Bugli et al., 2018; Tudose et al., 2019). However, functionalization of GO with molecules and chemical species do impact on the biological properties and may eventually reduce toxicity. Characterization of these functionalized GO shall include proper evaluation of the toxicity in relevant ex vivo model, so to identify the combinations showing adequate biocompatibility while maintaining the anti-microbial properties. Moreover, other GO-derived carbon-based nanomaterials such as for instance Quantum dots (De Maio, Palmieri, De Spirito, et al., 2019; Deng et al., 2022; Raja et al., 2022; Vasyukova et al., 2022) may provide more effective and less toxic scaffolds and overcome the observed limitations of the tested GO.” It could be as Conclusion.
We than the reviewer and we appreciate his/her suggestion. We have changed the revised manuscript accordingly.
- Figure 5. I don't see different between A, B, C picture, because description is the same, in text (results) we don't have any information it could be maturation in time or in what ? I don't understand part of this figure. The same situation is between E-G and I-K. May be it is my little experience in the similar experiments. And òòid the description of this Figure: (D, H, I) I is a mistake should be L
We thank the reviewer for the comment. Figure 5 represents the evaluation of cell activation following one hour GO treatment. Whereas in the columns (from left to right) we included untreated cell, GO (10µg/ml) and GO (100µg/ml)- stimulated cells; in the rows, we report CD4 lymphocytes, CD8 lymphocytes and monocytes, respectively. We have now revised all figure legends to better clarify each experimental setting and the relative result. We have also revised the results to avoid misunderstandings.
Reviewer 3 Report
The manuscript entitled Evaluation of the toxic activity of the Graphene Oxide in the ex vivo model of human PBMC infection with Mycobacterium tuberculosis is nicely written and well executed in terms of experimentation. However, the quality of manuscript need to be further strengthened by addressing the following queries:
1. In the abstract section, authors should avoid usage of abbreviations as it is not an essential requirement.
2. In the introduction section, the author should discuss the role of nanomaterials and compare their effects in the discussion section about the comparative efficacy of other nanomaterials.
3. In Figure 1 and 2, figure A and figure C represent the same information. Repetition may be removed.
4. In figure 3 D and 3 H, why the same controls have different values. Authors should discuss the variation in the control values in the manuscript.
General queries:
5. The font size of the manuscript is not uniform throughout the manuscript.
6. Reference format should be uniform as per the style of this journal of repute.
Author Response
Reviewer 3
The manuscript entitled Evaluation of the toxic activity of the Graphene Oxide in the ex vivo model of human PBMC infection with Mycobacterium tuberculosis is nicely written and well executed in terms of experimentation.
We thank the reviewer for the overall positive comments on the whole manuscript. In the point-by-point rebuttal, we will address the points raised by the reviewer.
However, the quality of manuscript need to be further strengthened by addressing the following queries:
- In the abstract section, authors should avoid usage of abbreviations as it is not an essential requirement.
We thank the reviewer for the observation. The abstract has been corrected accordingly.
- In the introduction section, the author should discuss the role of nanomaterials and compare their effects in the discussion section about the comparative efficacy of other nanomaterials.
We thank the reviewer for the suggestion. We have revised the introduction including the missing information on other carbon nanomaterials. In the revised manuscript, we have also improved the discussion adding information about GO effects on hPBMCs that may explain our results as well as may be related to our previous findings. Unfortunately, we can not compare our findings with other carbon nanomaterials, because of several differences in the experimental settings (cell lines, mycobacterial strain, nanomaterials features) that introduce a significant bias. We have underlined this point in this manuscript but also in our previous papers (see review De Maio et al, 2019; reference 6 in the main text).
- In Figure 1 and 2, figure A and figure C represent the same information. Repetition may be removed.
We thank the reviewer for the observation. Although quite similar, figure 1 and figure 2 show different experimental settings that reflect our previous paper (figure 1) and introduce to this one (figure 2). We have revised all figure legends to better elucidate our data and to avoid repetitions.
- In figure 3 D and 3 H, why the same controls have different values. Authors should discuss the variation in the control values in the manuscript.
We thank the reviewer for the comment. As explained in the text, panel 3D and 3H, report measurements obtained before debrisis and doublets removing and after their removing, respectively. For this reason, control values appear different.
General queries:
- The font size of the manuscript is not uniform throughout the manuscript.
We thank the reviewer for the suggestion. We have now revised the manuscript accordingly.
- Reference format should be uniform as per the style of this journal of repute.
We thank the reviewer for the suggestion. We have now revised the reference list accordingly.
Round 2
Reviewer 2 Report
Page 5 acapite:
Recently, Carbon-based Nanomaterials (CNM) like fullerenes, nanotubes, diamonds, graphite,
graphene and its conjugate Graphene-Oxide (GO), showed a broad direct antibacterial effect in in
vitro assays [Al-Jumaili et al., 2017; Bugli et al., 2018; Maas, 2016; Maleki Dizaj et al., 2015],
including activity against Mtb and other nontuberculous mycobacteria. GO-Ethambutol particles
inhibited M. smegmatis growth in axenic liquid culture [SAIFULLAH 1 e 2]; surface charged
fullerenes are able to inhibit the growth of M. avium and Mtb [BOSI]; GO in a reduced state exerts
antibacterial effect against Mtb [HAN]. It shall be noted that in many of these experiments
functionalyzed forms rather than a “pure form” of GO were used, with the functional groups
providing peculiar physical and chemical features [PALMIERI]. Interestingly, the investigators
What mean the names I highlighted (by the Reviewer, not in original version of the manusrcipt) in yellow?
REFERNCES:
In the version of the reviewed manuscript the references are not STILL in alphabetical order.
in the main text we don't have numbers of the references , so we don't know (when we go to the part References) which one i s the first, second etc. So the References must be in alphabetical order.
Author Response
Dear Reviwer,
We apologize for the alphabetical order of the names, they are now placed in numerical order. We also removed the highlighted portions of the changes to make the text more readable.